# DNA Methylation of Candidate Genes (ACE II, IFN-γ, AGTR 1, CKG, ADD1, SCNN1B and TLR2) in Essential Hypertension: A Systematic Review and Quantitative Evidence Synthesis

**DOI:** 10.3390/ijerph16234829

**Published:** 2019-12-01

**Authors:** Laurens Holmes, Andrew Lim, Camillia R. Comeaux, Kirk W. Dabney, Osatohamwen Okundaye

**Affiliations:** 1Nemours/A.I. DuPont Children’s Hospital, Nemours Office of Health Equity & Inclusion, 2200 Concord Pike, 7th Floor, Wilmington, DE 19803, USA; alim01@ufl.edu (A.L.); camillia.comeaux@nemours.org (C.R.C.); kirk.dabney@nemours.org (K.W.D.); okundayeit@gmail.com (O.O.); 2Biological Sciences Department, University of Delaware, Newark, DE 19711, USA; 3Institute of Public Health, Florida A&M University, Tallahassee, FL 32301, USA

**Keywords:** essential hypertension, DNA methylation and hypertension, hypertension candidate genes, epigenomics, socio-epigenomics

## Abstract

Physical, chemical, and social environments adversely affect the molecular process and results in cell signal transduction and the subsequent transcription factor dysregulation, leading to impaired gene expression and abnormal protein synthesis. Stressful environments such as social adversity, isolation, sustained social threats, physical inactivity, and highly methylated diets predispose individuals to molecular level alterations such as aberrant epigenomic modulations that affect homeostasis and hemodynamics. With cardiovascular disease as the leading cause of mortality in the US and blacks/African Americans being disproportionately affected by hypertension (HTN) which contributes substantially to these deaths, reflecting the excess mortality and survival disadvantage of this sub-population relative to whites, understanding the molecular events, including epigenomic and socio-epigenomic modulations, is relevant to narrowing the black-white mortality risk differences. We aimed to synthesize epigenomic findings in HTN namely (a) angiotensin-converting enzyme 2 (ACE II) gene, (b) Toll-like receptor 2 (TLR2) gene, (c) interferon γ (IFN-γ) gene, and (d) Capping Actin Protein, Gelosin-Like (*CAPG*) *gene*, adducin 1(ADD1) gene, (e) Tissue inhibitor of metalloproteinase 3 (*TIMP3*), (f) mesoderm specific transcript (MEST) loci, (g) sodium channel epithelial 1 alpha subunit 2 (SCNN1B), (h) glucokinase (CKG) gene (i) angiotensin II receptor, type1 (AGTR1), and DNA methylation (mDNA). A systematic review and quantitative evidence synthesis (QES) was conducted using Google Scholar and PubMed with relevant search terms. Data were extracted from studies on: (a) Epigenomic modulations in HTN based on ACE II (b) TLR2, (c) IFN-γ gene, (d) *CAPG*, ADD1, *TIMP3*, MEST loci, and mDNA. The random-effect meta-analysis method was used for a pooled estimate of the common effect size, while z statistic and I^2 were used for the homogeneity of the common effect size and between studies on heterogeneity respectively. Of the 642 studies identified, five examined hypermethylation while seven studies assessed hypomethylation in association with HTN. The hypermethylation of ACE II, SCNN1B, CKG, IFN-γ gene, and miR-510 promoter were associated with hypertension, the common effect size (CES) = 6.0%, 95% CI, −0.002–11.26. In addition, the hypomethylation of TLR2, IFN-γ gene, ADD1, AGTR1, and GCK correlated with hypertension, the CES = 2.3%, 95% CI, −2.51–7.07. The aberrant epigenomic modulation of ACE II, TLR2, IFN-γ, AGTR1, and GCK correlated with essential HTN. Transforming the environments resulting from these epigenomic lesions will facilitate early intervention mapping in reducing HTN in the US population, especially among socially disadvantaged individuals, particularly racial/ethnic minorities.

## 1. Introduction

Hypertension, which reflects elevated systolic and diastolic blood pressure, has been physiologically linked to elevated cardiac output, implying stroke volume, heart rate, and increased peripheral resistance, which is an obstacle to blood flow. Epidemiologic and animal studies have illustrated the role of a sedentary lifestyle, high fat diet, and sodium intake in predisposition to vascular constriction and volume loading. Additionally available translational epidemiologic data have implicated race in the complex and multifactorial etiologic pathway of essential hypertension, with blacks/African Americans (AA) being disproportionally affected [1]. For example, diet rich in the methyl group (CH3) such as processed red meat or stress as an environmental stimulus, results in elevated blood pressure via epigenomic mechanistic processes such as DNA methylation, histone acetylation, DNA hydroxymethylation, and DNA phosphorylation. Stress or social adversity induction in animal and human models have been shown to result in changes in the catecholamine pathway, implying the enzymatic regulation of blood pressure such as tyrosine hydroxylase and dopamine decarboxylase which are associated with no epinephrine and epinephrine elaboration vasoconstriction. Specifically, the DNA methylation of the promoter region of candidate genes involved in hemostasis and hemodynamics due to social stress, physical inactivity, or diet results in the inhibition of the gene transcription factor, impaired gene expression, and abnormal protein synthesis leading to disease development. For example, vasoconstriction occurs as the result of angiotensin I conversion to angiotensin II, a potent vasoconstrictor due to the elaboration of the angiotensin I converting enzyme (ACE I) [2]. Specifically, the up-regulation of the ACE II gene, due to the aberrant DNA methylation process, results in vasoconstriction, increased peripheral resistance, and hypertension (HTN).

Social signal transduction due to isolation, a social stressor, or discrimination reflects the fight or flight notion of the sympathetic nervous system (SNS) by the elaboration of norepinephrine and the beta adrenergic receptors activation, explicit in essential or primary HTN [3]. Adverse social environment such as racial discrimination, unstable social status, and psychosocial stressors serve as triggers of neural and endocrine activities, influencing the cellular response system and therefore, resulting in the activation of the intracellular signal transduction pathways and the subsequent repression or activation of transcription factors that are involved in the transcription of the gene bearing response element (GBRE). 

The DNA methylation mechanism of several candidate genes involved in this quantitative evidence synthesis (QES) require studies that have observed epigenomics signatures associated with the transcription factor inhibition at the 5 prime cytosine residue of the cytosine-phosphate-guanine (CpG) gene promoter region, resulting in the development of 5 methyl-cytosine (5mC) [4], implying DNA silencing without alternation in the DNA sequence. This process requires the excessive elaboration and availability of the DNA methyl-transferase that accelerates the process of methylation by recruiting the methyl group (CH3) from the S-adenosyl methionine (SAM) to the CpG island and shores on DNA. Additionally, the transcription factor inhibition influences the gene that codes for the RNA polymerase involved in the mRNA translation and subsequent gene expression and protein synthesis [5]. In general, DNA methylation involving 5mC correlates with gene repression or down regulation at the enhancer or promoter region. 

The pathophysiology of hypertension reflects the cardiac output, implying stroke volume and heart rate (myocardium contractility) as well as peripheral resistance, which infers that the obstacle to the flow of blood, the specific candidate genes in these pathways and their influence on the hypertension mechanistic process were examined and incorporated into the QES. In examining the peripheral resistance and its contribution to hypertension, the ACE II gene was assessed for hypermethylation. The function of this gene is to facilitate the conversion of angiotensin-I to angiotensin II, which is a vasoconstrictor, inducing hypertension. Specifically, the DNA methylation of this gene, which implies the impaired or inhibited transcription factors at the promoter region of the gene resulted in gene repression and angiotensin converting enzyme upregulation. 

The TLR2 (CD282, TIL4) identified as a Toll like receptor 2 gene, encodes for the Toll-like receptor (TLR) which is involved in pathogen recognition and innate and non-adaptive immune response activation. The TLRs activation by the pathogen-associated molecular pattern (PAMPs) results in the upregulation of signaling pathways in modulating a host’s inflammatory response [5]. Simply, the TLR2 plays a role in the inflammatory response, facilitating the immune system’s ability to recognize and respond to a pathogenic microbe or antigen. The epigenomic mechanistic process in TLR2 methylation involves the binding of the 5-cytosine at the promoter or enhancer region of the gene with the methyl group (CH3) due to the facilitating effect of the DNA methyltransferase and the subsequent inhibition of the transcription factor required for the gene product through the gene expression and subsequent impaired protein synthesis (inappropriate mRNA translation). The observed DNA methylation affects an individual’s response to stress, associated with an inflammatory response. Specifically, the methylation of the TLR2 may result in the up or down regulation of the gene product, implying an abnormal inflammatory response and the subsequent accumulation of pharmacologic mediators of inflammation, resulting in arteriosclerosis, arterial stenosis, plaque formation, and occlusion of blood flow (peripheral resistance). The cardiac compensation to peripheral resistance reflects increased cardiac contractility and the subsequent elevation of the cardiac output, leading to hypertension.

The interferon gamma (IFN-γ) are cell signals associated with increased elaboration, given viral infections an innate antiviral response. The insult to tissue or microbes colonization that results in antigenicity may provoke IFN-γ release and subsequent pharmacologic mediators of inflammation. The hypomethylation of this gene is indicative of the inhibition of this responses and the subsequent accumulation of the mediators of inflammation that could be caused by a stressful environment resulting in hypertension. 

The AGTR1 gene (angiotensin II receptor, type1) encodes for the angiotensin II receptor (AT1 receptor). The AT1 receptor is a protein involved in the renin-angiotensin system that regulates blood pressure (BP), fluid, and salt balance [6]. Specifically, this receptor acts as a vasopressor and regulates aldosterone secretion. This receptor binds with angiotensin II, stimulating chemical signals that result in vascular constriction and HTN. In addition, this binding results in aldosterone production, hence the increased renal absorption of salt and water with increased extracellular fluid resulting in BP elevation. The hypomethylation of the angiotensin II receptor has been assessed to predict hypertension. This protein molecule plays a role in binding with angiotensin II and the subsequent cellular function, namely vasoconstriction. The hypomethylation of AGTR 1 implies the lower availability of the methyl group at the CpG island with increasing 5-cytosine, resulting in a transcription factor activation and the subsequent gene expression due to the decreased availability of DNA methyltransferase. The consequence of this gene expression leads to the upregulation of the receptor and the availability of the angiotensin converting enzyme II binding, resulting in vasoconstriction. 

The ACE gene–angiotensin I converting enzyme encodes for angiotensin converting enzyme, which regulates BP and NaCl as well as H_2_O balance [7]. This enzyme cleaves Angiotensin I to Angiotensin II, a potent vasoconstrictor, elevating the BP. The ACE cleaves bradykinnin, a vasodilator, thus inactivating this molecule or protein, elevating the BP and hence the essential HTN. The protein encoded by this gene belongs to the angiotensin-converting enzyme family of dipeptidyl carboxydipeptidases and has a considerable homology to the human angiotensin 1 converting enzyme. This secreted protein catalyzes the cleavage of angiotensin I into angiotensin II. The organ- and cell-specific expression of this gene suggests that it may play a role in the regulation of cardiovascular and renal function, as well as reproductive process. 

Other candidate genes involved in HTN include: (a) SCNN1A (sodium channel epithelial 1 alpha subunit) gene, which encodes for the epithelial sodium channel (ENaC) complex. The ENaC transports sodium into cells and a decreased ENaC may result in excess volume or fluid in some organs. The mutation (deletion, restriction-shortening) in this gene had been observed in psuedohypoaldesterionism type 1 (PHA1), characterized by hyponatremia (low Na level), hyperkalemia (high K level), and severe dehydration. SCNN1B is the same as the alpha unit with impairment in this beta unit associated with sodium channel disruption and fluid balance as well as impaired Na reabsorption-hyponatremia. (b) TIMP3 gene (22q12.3) encodes for matrix metalloproteinases inhibitor proteins which are peptidases involved in degeneration of the extracellular matrix. The TIMP3 expression results from mitogenic stimulation while its mutation had been observed in autosomal dominant disorder namely, Sorsby’s fundus dystrophy. (c) GKG, glucokinase gene encodes a number of hexokinase family proteins. This gene encodes hexokinase phosporylate glucose to produce glucose-6-phosphate. CKG functions by providing G6P for the synthesis of glycogen. CKG mutation resulting in an alteration in enzyme activity is associated with diabetes and hyperinsulinemic hypoglycemia. (d) Adducin1 (ADD1) gene are cytoskeletal proteins encoded by three genes namely alpha, beta, and gamma, and binds with high affinity to Ca (2+)/Calmodulin [8]. (e) Keratin 13 (KRT 13) gene (17q21.2) encodes for the production of keratin protein-fibrous proteins that form the structural framework of epithelial cells. KRT13 gene partners with Keratin 14 (KRT4) gene (17q21.2) to form intermediate filaments, which functions by protecting the mucosa from being damaged by friction, or everyday physical stress [9]. The CAPG gene (2p11.2) encodes a member of the gelosin/villin family of actin regulatory protein. The encoded protein reversibly blocks the barbed ends of F-actin filaments in a Ca^2+^ and hence contributes to the control of actin-based motility in non-muscle cells [10]. 

Clinical and population-based data during the last three decades have implicated several biomarkers in HTN causal pathways. Recently, epigenomic studies explore the heritable changes to gene activity regulation unrelated to the DNA sequence, with these changes being rapid but reversible modifications and often in response to environmental changes [11]. 

Seven of the top ten leading causes of death in the US are attributable to chronic disease, which are influenced by epigenomic modulations. Cardiovascular diseases (CVDs) in the US remain the leading cause of mortality, accounting for 23% of all deaths, and has been linked in several studies to modifications such as hyper-and hypomethylation of phosphodiester-linked CpG sites or acetylation of histone proteins [12,13,14]. Hypertension, diabetes, hypercholesterolemia, and obesity continue to increase in the US by the age of 20 and older, despite reduction in smoking [12], which explains other predispositions to CVDs, including though not limited to adverse environment and gene interaction. The observed CVDs morbidity and mortality risk is highest among blacks/African Americans, males, in an advanced age, and individuals with low socio-economic status [12]. The cardiovascular risk which was age-independent observed differential mDNA profiles, with eight cytosine-phosphate-guanine (CpG) indicating differential mDNA patterns. These CpGs correlated with smoking and some were associated BMI (body mass index). In addition, the risk scores based on these mDNA patterns were related to CVDs outcomes and serve as predictive indices [13]. The majority of US annual healthcare expenditure (86% of $2.7 trillion) is on the treatment of chronic diseases [15]. The management of chronic disease has been a major focus of organizations including CDC (Centers for Disease Control and Prevention) and legislature such as the Affordable Care Act (ACA) in 2010. With the projected increasing number of Americans living with chronic conditions, a further understanding of chronic disease’s etiological pathways is fundamental to providing timely and effective care [16], requiring an evidence-based approach in addressing the cause of causes of HTN, namely social gradient and gene interaction. 

Within multifactorial models of disease, gene-gene interactions and gene-environment interactions via epigenomic modifications exist with varying degrees of effect and heritability [17]. The likelihood of engaging in health-promoting behaviors may themselves be epigenetically influenced [18]. Epigenomic modifications of the DNA can entail various chemical additions that alter the three-dimensional structural organization of the DNA, RNA, and proteins. The three most commonly examined modifications impacting chronic disease revolves around health-promoting behaviors [19]. The epigenomic mechanistic process that may influence cardiovascular pathologies that can influence HTN include: Methylation of CpG sites [20,21], varied functional groups to histone proteins [22], and binding of non-coding microRNA (miRNA) to target mRNAs [23,24].

Epigenomics remain a comparatively new field of study within genetics, computational biology, computer programing, environmental science, and public health. Studies on epigenomic pathways in disease causation could result in protective factors identification leading to healthy behaviors and improved outcomes in chronic disease, the leading cause of death in the US.

Current methods of epigenomic investigations are often limited to clinical trials or epigenome-wide association studies (EWAS) identifying individual modification sites, such as individual CpG methylation islands and miRNA sequences. These methods often yield inconsistent findings when analyzing epigenomic modulations or epigenomic signatures in chronic disease outcomes, particularly for specific sites and their individual roles in regulatory pathways. The current study aimed to assess the DNA methylation of candidate genes and miRNA binding in HTN for evidence-based data in informing intervention mapping, with the perspective of narrowing the subpopulation or racial/ethnic disparities in HTN incidence and mortality. 

## 2. Experimental Section

### 2.1. Design 

This study involved a systematic review and applied meta-analysis termed quantitative evidence synthesis (QES). This design was used to provide evidence on the implication of epigenomic alterations driven possibly by either social stress, isolation, diet, and physical inactivity in HTN development. 

The search included relevant literature identification with article selection based on study quality assessment performance. The qualitative synthesis of selected studies included data abstraction and synopsis. The QES included data extraction from eligible published literature, the pooled assessment estimation, the test for heterogeneity, and the creation of forest plots. 

#### 2.1.1. Design Description and Rationale

The overarching objectives of QES were to: (1) To minimize random error and (2) marginalize measurement errors, which have a substantial effect on the point estimate by down-drifting away from the null or towards the null. Since all studies have measurement errors and some studies have more measurement errors than others, QES assesses the differences between studies that are due to measurement errors. Additionally, because studies in medicine and public health are often conducted with small samples, such samples have increased random errors and hence restricted generalizability. The QES, which is a method of summarizing the effect across studies, increases the study or sample size and therefore minimizes random error and enhances generalizability of findings. Furthermore, QES integrates results across studies to identify patterns and to some extent establish causation. In effect, the overall relevance of QES is to generate scientific data that are cumulative and reliable in improving health or other conditions upon which it is applied. 

The methodology used in QES differs from that of traditional meta-analysis. While meta-analysis utilizes fixed and random effect methods, QES only employs the random effect method and examines heterogeneity after and not before the pool estimates. The fixed effect method is only applicable to QES when the combined studies or publications are from a multi-center trial where the study protocol is identical. However when studies are combined from different settings, observation and measurement errors induce significant variability in the observed estimates, limiting such combination without adjusting for between studies variability. The random effect method compensates for the between studies variability, hence its unique application in QES. 

Scientific endeavor makes sense of the accumulating literature in medicine and public health, given confounding and contradicting results. QES reflects such attempts at study integration for public health and clinical decision-making. A unique feature of QES is temporality, in which findings in QES accumulate with time. For example, if QES was performed on the implication of epigenetic modification, such as DNA methylation and histone acetylation, with respect to hypertension development, this study must identify time of conduct and continue to add findings and reanalyze the data for contrasting or negative findings with time. Subsequently, the emergence of new data on epigenomic modification, gene expression, and post-translation histone acetylation or methylation has an impact in changing the results of QES and moving evidence in a different direction. 

Science and scientific endeavors are not static but dynamic, implying evidence transformation following the emergence of new data. The scientific community cannot wait until evidence accumulates to such a point that no further addition is required with respect to evidence discovery, in order to initiate an intervention regarding epigenomic intervention mapping for specific risk characterization in predisposing factors associated with HTN. Consequently, QES can inform and provide at any point in time the knowledge required in order to understand the disease process, characterize risk, improve prognosis, as well as control and prevent disease at patient and population levels. 

#### 2.1.2. Search Engines and Strategies

We conducted an online database search of Google Scholar and MEDLINE via PubMed in June 2018. Search terms were created based on medical subject headings (MeSH) and terms used in epigenomic literature reviews of HTN in order to maximize sensitivity: (gene expression OR DNA methylation OR histone acetylation OR microRNA OR gene transcription OR mRNA OR histone methylation OR epigenotype) AND hypertension (angiotensin-converting enzyme, C-reactive proteins, fibrinogen, plasminogen activator inhibitor I, aldosterone renin, B-type, natriuretic peptide, homocysteine, N-terminal proatrial natriuretic peptide, catecholamines pathways, enzymes involved in catecholamines pathways, dopamine decarboxylase, tyrosine hydroxylase) OR (gene expression OR DNA methylation OR histone acetylation OR microRNA OR gene transcription OR mRNA OR histone methylation OR epigenotype) AND hypertension (angiotensin-converting enzyme, C-reactive proteins, fibrinogen, plasminogen activator inhibitor I, aldosterone renin, B-type, natriuretic peptide, homocysteine, N-terminal proatrial natriuretic peptide, catecholamines pathways, enzymes involved in catecholamines pathways, dopamine decarboxylase, tyrosine hydroxylase) AND epigenetic modification (gene expression OR DNA methylation OR histone acetylation OR microRNA OR gene transcription OR mRNA OR histone methylation OR epigenotype) AND (hypertension OR angiotensin-converting enzyme OR C-reactive proteins OR fibrinogen OR plasminogen activator inhibitor I OR aldosterone renin OR B-type OR natriuretic peptide OR homocysteine OR N-terminal proatrial natriuretic peptide OR catecholamines pathways OR enzymes involved in catecholamines pathways OR dopamine decarboxylase OR tyrosine hydroxylase) OR (gene expression OR DNA methylation OR histone acetylation OR microRNA OR gene transcription OR mRNA OR histone methylation OR epigenotype) AND (hypertension OR angiotensin-converting enzyme OR C-reactive proteins OR fibrinogen OR plasminogen activator inhibitor I OR aldosterone renin OR B-type OR natriuretic peptide OR homocysteine OR N-terminal proatrial natriuretic peptide OR catecholamines pathways OR enzymes involved in catecholamines pathways OR dopamine decarboxylase OR tyrosine hydroxylase) AND epigenetic modification.

Additionally, we performed hand searches through reference lists of relevant articles. Such articles were identified in advance based upon their investigation of epigenomic modifications on hypertension (Figure 1). 

#### 2.1.3. Eligibility Criteria 

Eligible articles had to meet the following criteria: Study published in English from 1 January 2000 to 1 June 2018, study investigates hypertension and epigenomic changes, study with a well-defined outcome, namely hypertension, and study contains quantitative data, such as the parameter values (odd ratio, risk ratio, relative risk). Studies with a loss to follow up >25% or sample size smaller than 10 were excluded to lessen the likelihood of selection bias and sparse data bias, respectively. Inclusion criteria were developed in order to maximize an inclusion of any potentially useful findings while limiting the inclusion of irrelevant data. When a study was identified alluded to the existence of quantitative data but did not disclose any of it in the article itself, we contacted the authors via e-mail in an effort to obtain additional data. One author (A.L.) screened abstracts for inclusion in the full-text evaluation.

Two authors (A.L. and O.O.) independently read the full texts and extracted the data into a QES data sheet, with kappa = 0.98, 98%, implying strong or high inter-rater reliability. Discrepancies in agreement were resolved through a discussion between the study investigators. The final list of studies included in the qualitative and quantitative syntheses was the product of this discussion following the initial independent review. Studies included in the qualitative synthesis had to meet the inclusion criteria above. Studies included in the QES had to meet the inclusion criteria, implying studies with epigenomic modification measured by either DNA methylation or histone acetylation and hypertension. 

#### 2.1.4. Study Variables 

The study variables included essential hypertension as the response variable while epigenomic alterations as well as biomarkers of hypertension as independent variables. Other study variables included age and sex. 

#### 2.1.5. Data Collection 

Data was collected from all eligible studies based upon the outcome variables and the specific research questions. For the qualitative synthesis, all available data concerning hypertension and epigenomic markers were obtained in the data collection strategy. For the QES, we abstracted data on the proportion of those with and without hypertension given epigenomic modification. Where data were not available on the measure of the outcome, we estimated that based on the absence or presence of epigenomic changes and the outcome being hypertension. 

#### 2.1.6. Study Quality Assessment 

Two authors (A.L. and O.O.) assessed the eligible studies’ quality based upon the study design, sampling techniques, hypotheses, clarity of aims or purposes, and adequacy of statistical analysis. Studies were also assessed for any confounding factors that might have influenced the outcomes and any potential bias, including selection, information, and misclassification biases. The study quality assessment technique was in line with the preferred method of reporting for systematic reviews (PRISMA statement).

### 2.2. Statistical Analysis 

Descriptive or exploratory analysis was performed to examine qualitative scale measurement data for frequencies and percentages. The inferential statistics, namely QES analysis that involved the pooled estimate or the common effect sizes, was performed prior to the heterogeneity test. In addition, we created a template to transform the proportion of epigenomic modification and hypertensive cases into percentage, standard error, and 95% confidence intervals, to enable the application of the meta-analytic command using STATA, namely *metan percent lowerci upperci, label (namevar = study) random*.

To test the hypotheses with respect to the implication of epigenomic modification in hypertension, we used the random-effect analytic method of DerSimonian–Laird [25]. The DerSimonian–Laird method was applicable given significant studies on heterogeneity. This procedure, namely the random-effect method, examined the between-studies effect as well as the effect sizes of the combined studies or the common effect size in relation to the individual effect sizes, weighing each study for their contribution to the overall sample size involved in the pooled estimate. In addition, the common effect size precision was measured by the 95% confidence interval. The effect size heterogeneity was estimated by testing the null hypothesis that the common effect size = 0 based on the standardized normal (z statistic). Additionally, a meta-regression was performed to assess the subgroup effects of the candidate genes in either hypo or hyper DNA methylation in essential or idiopathic HTN. This process allowed for the examination of potential confoundings in applied meta-analytic designs, facilitating the subgroups heterogeneities in result interpretations involving the overall, individual, and subgroup effect sizes. 

The heterogeneity test was performed to determine variability among studies based on the individual studies effect sizes in relation to the common effect size (diamond). Using Q = (1/variance_i) × (effect_i − effect_pool)^2^ we determined the heterogeneity in this QES. The variance was estimated using: Variance_i = ((upper limit − lower limit)/2 × z))^2^. The test of heterogeneity reflected the variation in the effect size (ES) that is attributable to the differences between studies’ effect sizes. The significance level was set at 5% (0.05 Type I error tolerance) and all tests were two tailed. The entire analyses were performed using STATA 15.0 (StataCorp, College Station, TX, USA).

## 3. Results and Discussion

### 3.1. Results

These data represent epigenomic aberrations in essential hypertension from the perspective of predisposition, given gene-environment interaction that characterizes epigenomic modulation or epigenomic lesions of the candidate genes involved in hemodynamics and homoestatsis. The epigenomic moduation reflects on-going changes within the gene promoter region that does not affect the DNA sequence, but alters the transcription and the subsequent gene expression such as repression. The impairment in this modulation, termed epigenomic aberration or lesion, induces alteration in plasticity resulting in a decreased response to cellular damage and inflammation. Specifically, such alteration affects hemodynamics and hemostasis, resulting in sodium and fluid imbalance and the subsequent HTN. 

We examined studies on DNA methylation that involved the methylation of the gene enhancer or promoter region, mainly the CpG islands, which affects transcriptional activities as well as the transcription factor’s inability for mRNA to translate and consequently express the gene, impacting protein synthesis (Table 1). Such limitations or inabilities eventually results in abnormal protein synthesis as initially observed implying a disease process, impaired prognosis, and subsequent mortality. Since hypertension involves several biomarkers resulting in vasoconstriction as reflected in increased peripheral resistance, increased stroke volume, and increased heart rate the genes associated with these biomarkers were the main focus of this aberrant epigenomic modulation in HTN. 

#### 3.1.1. DNA Hypo-Methylation of HTN Candidate Genes

With DNA methylation the most commonly used epigenomic mechanistic process and current epigenomic detection technology, namely bisulfite pyrosequencing, we examined published literature for either hypo- or hyper-methylation in order to observe patterns and identify epigenomic causal pathways or association in HTN. Figure 2 presents the overall or common effect size of DNA hypo-methylation of the candidate genes namely ACE2, IFN-γ, TLR2, SCNN1A/1B, GCK, ADD1, and AGTR1) involved in HTN, as well as the meta-regression of these individual or specific genes. The mechanism of epigenomic modulation in these studies solely involved the DNA methylation of the cytosine-phosphate-guanine (CpG) 1, 4, 6, and 8, with respect to the TLR2 gene as well as the promoter region of ACE2, mainly CpG2 and CpG 5. Seven studies observed DNA hypo-methylation as an exposure function of hypertension. The sample size for these studies was 1105, while the mechanism of epigenomic modulation was the DNA methylation at the CpG islands by bisulfite pyrosequencing. The pooled estimate (the common effect size) for the hypomethylation indicated a common effect size (CES) = 2.3%, 95%, CI, −2.51–7.07. The variabilities between the common effect size and the individual effect seizes was estimated, I^2 = χ^2^ (7) = 107, 54.5, *p* < 0.001, indicative of substantial variances. The observed CES was significantly different from zero (0), z = 3.87, *p* < 0.001, negating the null hypothesis of a zero common effect size.

#### 3.1.2. Dense DNA Methylation of HTN Candidate Genes

In Figure 3, the forest plot demonstrates five studies that observed hypermethylation, implying the dense DNA methylation of hemostasis and hemodynamics genes namely ACE2, SCNN1B, IFN-γ, and CKG, as an exposure function in HTN causation and prognosis. The hypermethylation ranged from 0.65% to 16.0% with respect to the individual effect sizes in the forest plot. The sample size for the hypermethylation was 1105, implying a reasonable study size for a statistically stable finding of the effect of DNA methylation on the HTN causal pathway. The combined or pooled summary estimates for hypermethylation, although imprecise in terms of precision parameters, indicated a substantial common effect size (CES) = 6.0%, 95% CI, −0.002–11.26. The variabilities between the common effect sizes and the individual effect sizes was estimated, heterogeneity (I^2) = χ^2^ (10) = 2610.3, *p* < 0.001. The observed common effect size was significantly different from zero, z = 11.96, *p* < 0.001, negating the null hypothesis of a zero common effect size.

### 3.2. Discussion

Hypertension as a chronic disease involving cardiovascular system, mainly cardiac output and peripheral resistance, is associated with several biomarkers, genes, and gene products. However subpopulations genetic heterogeneity does not explain the racial/ethnic differences in essential HTN incidence, prognosis, and mortality. A possible explanation of racial/ethnic, sex, and age differences in this manifestation is provided in greater part by the hemostasis and hemodynamics genes and endogenous as well as the exogenous environment, which includes social adversity, social isolation, diet, physical activities, as well as persistent stress interaction. The subpopulation differences in exposure to objective and subjective isolation, low SES, and unstable social status facilitates our understanding of alteration in neural activities that result in vasoconstriction and subsequent increased peripheral resistance. With recent advances in epigenomic modulations in disease causation, prognosis, and survival, there is an urgent need for evidence-based data on further understanding of the biologic mediation of psychosocial etiologies of HTN via the gene as well as interaction between physical, chemical, and social environments. 

The purpose of the current study was to examine published literature on DNA methylation as the most commonly utilized epigenomic mechanistic process in HTN causal pathways, implying that DNA methylation of the candidate genes involved in HTN were biomarkers of causation and prognosis. An applied meta-analytic design termed quantitative evidence synthesis (QES) was utilized with the intent to provide scientific evidence on the influence of mDNA on gene expression and the subsequent under-expression or upregulation of genes involved in cardiac output and peripheral resistance. There are a few relevant findings from this QES. First, biomarkers of homeostasis and hemodynamics were identified and their gene correlates. Secondly, DNA methylation of the candidate genes in cardiac output (stroke volume and heart rate) and peripheral resistance were observed in HTN causal pathways. Thirdly, DNA hyper-methylation of ACE, ACE2, SCNN1B, IFN-γ, and CKG was observed in essential HTN. Fourthly, DNA hypomethylation of ACE2, IFN-γ, TLR2, SCNN1A/1B, GCK, ADD1, and AGTR1 correlated with HTN.

There are several postulated risk factors in HTN, implying a multifactorial etiology as well as environmental differences in HTN predisposition. The postulated risk factors in HTN include race, sex, diet, medication, drugs, alcohol, high sodium ingestion, smoking, a sedentary life style, overweight/obesity, stress, social isolation, and physical inactivity, as well as a family history of HTN. The subpopulation variances in lifestyle and living conditions interaction with the genes, as observed in epigenomic modulation provided an additional understanding in HTN risk and predisposition. While some of these risks have been established in HTN, risk synergism is more pronounced in subpopulations or individuals with aberrant epigenomic modulation of the candidate genes involved in hemodynamics. 

We demonstrated that ACE gene upregulation increased the risk of HTN, given its role in vasoconstriction and sodium/water imbalance. This affirmation supports previous literature on sodium loading, fluid retention, and decreased volume depletion in HTN [26]. The DNA methylation of ACE has a biologic underpinning in converting angiotensin I to angiotensin II, a potent vasoconstrictor, resulting in increased peripheral resistance and arterial stenosis, leading to an elevated blood pressure or hypertension (HTN). The DNA hypo-methylation was observed in IFN-γ gene, implying innate antiviral response inhibition, which increased the inflammatory process and mediators, thus enhancing arterial plague, arteriosclerosis, arterial stenosis, and HTN. This explanation is supported by studies that implicate chronic inflammation in HTN [27]. In contrast a study observed hypermethylation of the interferon gamma gene. Similarly, the DNA hypo-methylation of the SCNN1B gene may result in sodium channel disruption and fluid imbalance. Depending on the dysregulation, sodium channel disruption may induce hypo or hypertension. Additionally, the DNA hypo methylation of AGTR1 gene may predispose to hypertension by increasing the receptor for the potent vasoconstrictor, angiotensin II.

The observed DNA hypomethylation of TLR2 CpG1 and GCK CpG4 in HTN [28] is indicative of the inherent inability of this gene to elaborate the inflammatory cytokine in response to tissue damage, with such cellular insufficiencies resulting in vascular damage and microvascular compromization signaling disrupted homeostasis and hemodynamics. The observed aberrant epigenomic modulation of the TLR2 island is supported by studies that implicated depression in the pathogenesis of HTN. The hypomethylation of TLR2 cytosine-phospate-guanine (CpG) results in accumulated imflammation, arterial stenosis, and subsequent elevated or increased peripheral resistance. The observed TLR2 CpGs hypomethylation directly correlated with IL-6 gene hypomethylation. The hypomethylation of IL-6, CpG2, and CpG3 induced inflammation and endothelial dysfunction, elevating blood pressure. This epigenomic modulation of IL-6 gene affirms the implication of smoking, alcohol, and gender in HTN predisposition, where hypomethylation of IL-6, CpG2, and CpG3 was observed in males, smoking, and alcohol [29]. The hypo methylation of ADD1, implying aberrant epigenomic modulation results in upregulation and increased expression of α-adducin, leading to increased activities of the Na+-K+pump, inducing sodium reabsorption, volume expansion, and subsequent HTN. The observed inverse correlation between SCNN1B DNA methylation and essential HTN, implying hypomethylation, may be due to the upregulation of the protein expression of SCNN1B, thus increasing the function and activities of the epithelial sodium channel (ENaC) and enhancing Na+ reabsorption and fluid or water retention.

This QES illustrated that the hypermethylation of some genes, including angiotensin 1 converting enzyme 2 (ACE2) gene at promoter CpG4 and CpG5, directly correlated with hypertension. Available data observed DNA hypermethylation in general to suppress gene transcription, resulting in gene silencing [30]. Overall mDNA influenced protein-DNA interaction, gene expression, chromatin structure, and genome stability. Specifically, the hyper-methylation of CpG 4 and CpG5 of the ACE2 promoter may result in renin-angiotensin system (RAS) dysregulation. With the observed dense DNA methylation of ACE2, implying transcription inactivation, and the AGTR1 gene upregulation, there appeared to be RAS imbalance and subsequent vasoconstriction, resulting in essential HTN.

Although mDNA at CpG inhibits or represses transcription, it is not fully understood if exons methylation of most genes inversely or directly correlates with transcriptional activities, despite the observed inverse correlation between NR3C1 1F exon methylation in major depressive disorders [31]. While several methods had been utilized for epigenomic modulation detection, the current method involves bisulfite sequencing. With this process, the unmethylated cytosine residues were converted to uracil via the application of sodium bisulfite and alkaline treatment, while the methylated cytosine remain unconverted, and thereafter the bisulfite-treated was sequenced to identify the methylated cytosine [31]. Whereas aberrant epigenomic modulations remain a pathway to disease etiology, prognosis, survival, and morality as well as subpopulation differences in health outcomes, more investigations with high potentials for causal inference are required in this trajectory for effective intervention mapping and treatment effect homogeneity.

Despite the strength of this study in implicating gene and environment interaction in HTN predisposition, there are some limitations. First, QES is a retrospective study, implying the potentials for information, selection, and misclassification biases. Secondly, as a literature review, implying studying studies prior to scientific statement generation and evidence-based data on HTN causation or association, there is a potential for unmeasured confounding in the studies that constitute this QES. Thirdly, because of the design with its sample and sampling technique, patient and bioassay heterogeneity, there is potential for reverse causation in the observation of the DNA hyper- and hypo- methylation of the candidate genes involved in HTN. Specifically since the individual studies that constitute this QES are non-experimental epidemiologic designs, mainly case-control, cautious optimism should be applied in the causal inference application of these findings. Further, epigenomic modulation is reversible and the timing of the collection of the specimen is essential in such investigations, implying caution in the application of this QES in intervention mapping or specific risk characterization in HTN prevention and control.

## 4. Conclusions

In summary, dense DNA methylation of ACE, ACE2, SCNN1B, IFN-γ, and CKG correlated with essential HTN, as well as DNA hypo-methylation of ACE2, IFN-γ, TLR2, SCNN1A/1B, GCK, ADD1, and AGTR1 in HTN. The observed DNA methylation played a role in hemodynamics imbalance, leading to abnormal cardiac output and increased peripheral resistance, hence essential, primary, or idiopathic HTN. Epigenomic investigations reflecting the detection of aberrant modulation involved the assessment of hereditable changes in gene expression that occurred in the absence of underlying DNA sequence as observed in epigenomic mechanistic process namely: DNA methylation, histone protein modifications, phosphorylation, microRNA, and DNA microarray. Unlike DNA or basic inherited structure remaining constant during ontogeny, epigenomic codes or programming undergoes dramatic modification during embryogenesis, reflecting differential patterns in gene expression and the related tissue development and cellular function. These epigenomic signatures although transgenerational are reversible, implying societal collective effort and responsible action in changing the environment for the socially disadvantaged individuals and groups thus, reducing black-white risk differences in HTN incidence and mortality in the US.

## Figures and Tables

**Figure 1 ijerph-16-04829-f001:**
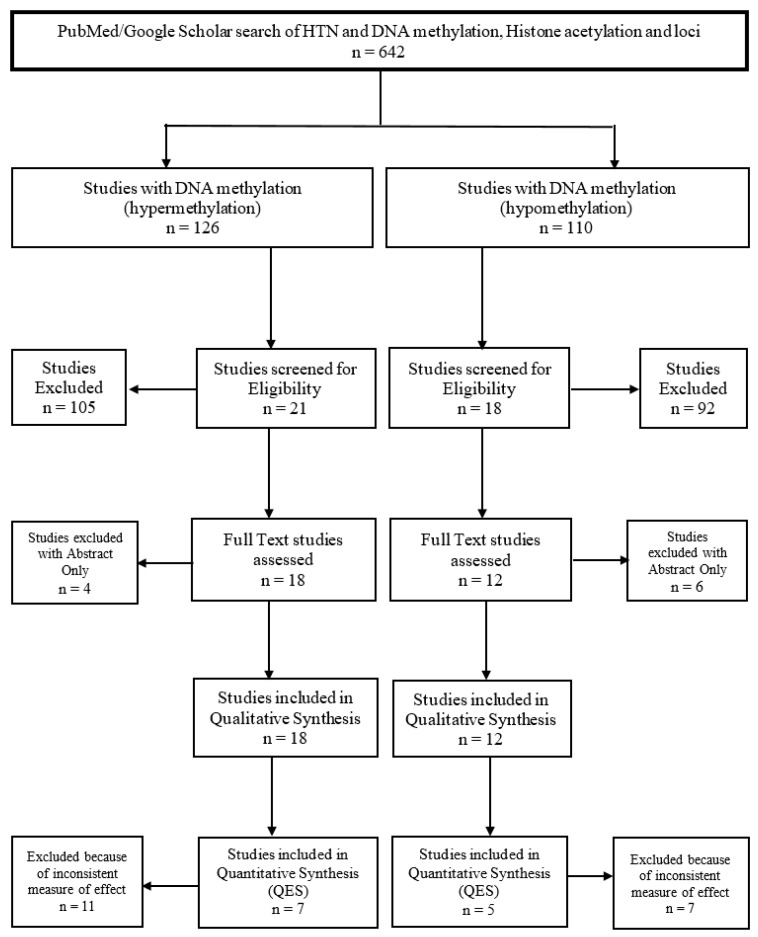
DNA methylation and hypertension (HTN).

**Figure 2 ijerph-16-04829-f002:**
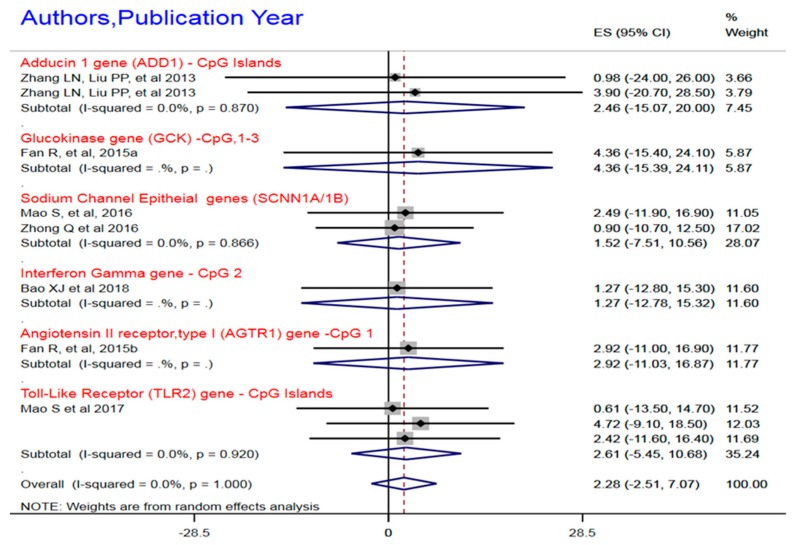
Hypomethylation of candidate genes in essential hypertension. Notes and abbreviations: Random effect meta-analysis with meta-regression for sub-gene common effect size estimation. cytosine-phosphate-guanine = CpG. The heterogeneity test for a single gene mDNA is not quantified and represented as .%, and *p* = .

**Figure 3 ijerph-16-04829-f003:**
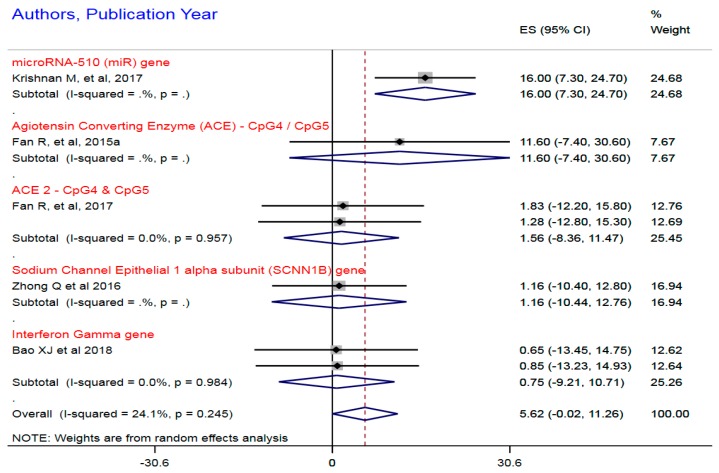
Hyper or dense methylation of candidate genes in essential hypertension. Notes and abbreviations: Random effect meta-analysis with meta-regression for sub-gene common effect size estimation. cytosine-phosphate-guanine = CpG. The heterogeneity test for a single gene mDNA is not quantified and represented as .%, and *p* = .

**Table 1 ijerph-16-04829-t001:** DNA methylation (mDNA) process and selective candidate genes in essential HTN.

Author, (Year), Study	Specimen/Source/Sample	Methylation Process	Patient/Subject Characteristics	Methylation Profile/ Status
Mao, T., et al. (2017) TLR2 Hypo-mDNA in e-HTN	Blood sample–antecubital vein	Bisulfite pyrosequencing of TLR2 CpGs.	96 controls and 96 incident essential HTN cases	Hypo-methylation and increased transcription
Mao, S., et al. (2016) SCNN1A Dense mDNA in e-HTN	Peripheral blood sample	Sodium bisulfite pyrosequencing technology of 6 CpG dinucleotides of SCNN1A.	60 incident and 60 prevalent cases and 60 comparable controls	Hyper-methylation, 16% incident case and 15% prevalent case relative to controls
Zhang, L., et al. (2013) ADDI Hypo-mDNA in e-HTN	Overnight fasting-peripheral blood sample	Bisulfite pyrosequencing of α-adducin (ADD1) gene of 5 CpGs promoter	33 essential HTN (14 males and 19 females) and 28 controls (14 males/females)	Hypo-methylation with sex differential and stable findings in males
Zhong, Q., et al. (2016) SCNN1B Hypo-mDNA in e-HTN	Peripheral blood sample	Bisulfite pyrosequencing of SCNN1B gene at 6 CpG sites	98 controls, 94 incident and 94 prevalent HTN cases	DNA hypomethylation, inverse correlation with HTN
Fan, R., et al. (2017) ACE2 Dense mDNA in e-HTN	12 h overnight fasting blood sample-antecubital vein from	Bisulfite pyrosequencing of ACE2 at 5 CpG inucleotides (1–5)	96 patients with essential HTN and 96 comparable controls	Dense DNA methylation–effect of sex on methylation profile
Bao, X.J., et al. (2018) IFN-γ gene in e-HTN	peripheral blood DNA	Bisulfite pyrosequencing of IFN-γ gene of 6 CpG sites	96 cases of HTN and 96 comparable controls	Hypo-methylation

**Notes and abbreviations:** e-HTN = essential hypertension; CpG = cytosine-phosphate-guanine; INF-γ = interferon gamma; mDNA = DNA methylation, TLR2 = Toll like receptor 2, SCNN1A = sodium channel epithelial 1 alpha subunit, cytosine-phosphate-guanine, specific site 3 (CpG3).

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
