# Peer review of "DNA Methylation of Candidate Genes (ACE II, IFN-γ, AGTR 1, CKG, ADD1, SCNN1B and TLR2) in Essential Hypertension: A Systematic Review and Quantitative Evidence Synthesis"

_ijerph, 2019, doi:10.3390/ijerph16234829_

Round 1
Reviewer 1 Report
This study is an interested systematic review based on qualitative evidence synthesis (QES) approach of candidate genes promoter DNA methylation in idiopathic hypertension. In my opinion, this study is of interest with novelty aspects but needs significant revision before accepted for publication. Firstly, presented results of aberrantly methylated gene promoters and estimated ES (in the text and Figures 2 and 3) are equivocal. Please, verify gene abbreviations (Figures, Conclusion) and ES values presented in the Abstract and Results/Figures. Secondly, references of 12 examined studies and basic characteristics (number of cases and controls, gender, ethnicity etc.) should be included. Thirdly, I suggest to clearly mention that epigenetic features were studied in the blood.
Author Response
Reviewer I
Reviewer’s Comment: This study is an interested systematic review based on qualitative evidence synthesis (QES) approach of candidate genes promoter DNA methylation in idiopathic hypertension.
Authors’ Response: Thank you very much for the observation and the related comment. We acknowledged in this study the multifactorial etiology of hypertension (HTN) and the inherent challenges in response to therapy upon diagnosis. The gene and environment interaction in HTN predisposition allows for specific risk characterization and therapy induction prior to the standard of care.
Reviewer’s Comment: In my opinion, this study is of interest with novelty aspects but needs significant revision before accepted for publication. Firstly, presented results of aberrantly methylated gene promoters and estimated ES (in the text and Figures 2 and 3) are equivocal.
Authors’ Response: Thank you very much for this observation. We have very carefully reviewed the abstract and the result section and have identified the errors and rectified them, thus communicating this findings to the scientific community with clarity and consistency.
Reviewer’s Comment: Please, verify gene abbreviations (Figures, Conclusion) and ES values presented in the Abstract and Results/Figures.
Authors’ Response: Thanks, we have explained and carefully described the genes throughout the paper, given that gene notions and abbreviations could be confusing especially among those who are not experts in genetic or epigenomic studies but are interested in utilizing such findings in enhancing patient care, counselling and disease prevention.
Reviewer’s Comment: Secondly, references of 12 examined studies and basic characteristics (number of cases and controls, gender, ethnicity etc.) should be included.
Authors’ Response: Thanks for the comment! We have addressed this by identifying the populations at risk for CVDs as well as epigenomic modulations with isolated risk factors in HTN and CVDs. With such clinical and epidemiologic correlates at the introduction, readers are more likely to observe value in this paper.
Reviewer’s Comment: Thirdly, I suggest to clearly mention that epigenetic features were studied in the blood.
Authors’ Response: thanks for this methylation process comment. We have very carefully addressed this. Please see table 1.
Reviewer 2 Report
Holmes et al reviewed literature about the role of DNA methylation on genes involved in essential hypertension, including ACE II, IFNγ, AGTR1, and TLR2. Authors speculate about the plasticity of epigenomics signatures as sensors of environmental changing and social disadvantage. This speculation is used by Holmes and co-workers to possibly explain the differences in hypertension incidence and mortality in US between Caucasian and blacks/African Americans.
The present literature revision sounds interesting, however some aspects needs to be better explained and clarified.
Sentences in lines 20-25 and 97-101 are too long; Sentences in lines 39-41; 111-113; 120-121; 173-174; 210-212; 219-222; 223-224; 397-398; 426-428 should be rephrased; Could authors motivate why they just focus on histone acetylation and not on histone modifications in general? Could authors list examples related to epigenomic mechanistic processes mentioned at line 55? Authors suggest that social signal transduction might influence epigenomics signature explaining differences in hypertension incidence and mortality according ethnics and life style. Is not clear how authors analyzed the influence of social signals on epigenomics. Did authors check the epigenomic status of genes involved in pathways described in line 67-74? Further, is not clear how racial and ethnic features influence epigenomics. Results are not reported according to racial and ethnic features. References at page 3 are missing. The statement in line 159-161 is not correct. Epigenetics is a discipline studied since long time. Authors need to improve resolution of all figures. The present manuscript sometime mentions hemodynamics and homeostasis and some others hemodynamics and hemostasis, which one is correct?
In the conclusion section, authors mention prions (line 440). This is the first time prions are mentioned in the text, could the authors explain this statement?
Author Response
Dear Editor,
Thank you and the reviewer for the opportunity to revise and resubmit our manuscript for a reconsideration of publication. Below please find our response to reviewer's comment:
Reviewer II
Reviewer’s Comment: Holmes et al reviewed literature about the role of DNA methylation on genes involved in essential hypertension, including ACE II, IFNγ, AGTR1, and TLR2. Authors speculate about the plasticity of epigenomics signatures as sensors of environmental changing and social disadvantage. This speculation is used by Holmes and co-workers to possibly explain the differences in hypertension incidence and mortality in US between Caucasian and blacks/African Americans.
The present literature revision sounds interesting, however some aspects needs to be better explained and clarified.
Authors’ response: Thanks for the interesting and very meaningful observation of the motivation of this applied meta-analysis (QES) study. We have very carefully reviewed your comments and suggestions and have addressed this given the feasibility and the current state of data in the field.
Reviewer’s comment: Sentences in lines 20-25 and 97-101 are too long; Sentences in lines 39-41; 111-113; 120-121; 173-174; 210-212; 219-222; 223-224; 397-398; 426-428 should be rephrased; Could authors motivate why they just focus on histone acetylation and not on histone modifications in general?
Authors’ response: Thanks for the observation. We have revised these sentences for clarification. Also studies with DNA methylation were used in this QES and not histone acetylation. Please see table 1. However as epigenomic mechanistic process, we did mention these processes.
Reviewer’s comment: Could authors list examples related to epigenomic mechanistic processes mentioned at line 55? Authors suggest that social signal transduction might influence epigenomics signature explaining differences in hypertension incidence and mortality according ethnics and life style.
Authors’ response: Thanks for the suggestion. We have mentioned epigenomic mechanism for aberrant detection. The social signal transduction mentioned in the paper had been observed and is referenced in the recent paper by author on NR3C1 gene and epigenomic modulation as well as the provocation of CTRA gene in leukocytes, IL-6 and IFN-y.
Reviewer’s comment: Is not clear how authors analyzed the influence of social signals on epigenomics. Did authors check the epigenomic status of genes involved in pathways described in line 67-74? Further, is not clear how racial and ethnic features influence epigenomics.
Authors’ response: Thanks for the observation. The observation is not part of the study, and there is no such analysis in this study. Only genes involved volume balance implying cardiac output and peripheral resistance were examined for the CpG aberrant modulations. Race and ethnicity are surrogates to SES, living conditions, educations, discriminations, social stressors, food insecurity, etc. These enviroments interact with human genes at the CpGs inducing silencing, transcription inhibition and impaired gene expression.
Reviewer’s comment: Results are not reported according to racial and ethnic features. References at page 3 are missing. The statement in line 159-161 is not correct.
Authors’ response: Thanks! The study was not designed to observe racial and ethnic difference in mDNA associated with HTN. We have reviewed this page and observe it to be appropriate with respect to referenced literature in this introduction.
Reviewer’s comment: Epigenetics is a discipline studied since long time.
Authors’ response: Thanks for the observation. Epigenetic studies are not novel but the genomics that set the platform for epigenomic is still at its infantile stage as there are several missing links in this field today. In effect, wherever epigenetic is used in this paper , we recommend it to be changed to epigenomics.
Reviewer’s comment: Authors need to improve resolution of all figures.
Authors’ response: Thanks for the observation from the forest plots. We have addressed this by increasing the resolution.
Reviewer’s comment: The present manuscript sometime mentions hemodynamics and homeostasis and some others hemodynamics and hemostasis, which one is correct?
Authors’ response: Thank you for this observation and the comment therein. These terms are used to indicate fluid balance and imbalance, as volume loading and depletion is used to explain HTN, the application of these two terms is very appropriate.
Reviewer’s comment: In the conclusion section, authors mention prions (line 440). This is the first time prions are mentioned in the text, could the authors explain this statement?
Authors’ response: Thanks for the observation. We have addressed this by avoiding the mention of prions in this QES.
Round 2
Reviewer 1 Report
The authors have responded to all my remarks and made the necessary changes to the manuscript.